# Distributed Acoustic Sensing of Strain at Earth Tide Frequencies

**DOI:** 10.3390/s19091975

**Published:** 2019-04-27

**Authors:** Matthew W. Becker, Thomas I. Coleman

**Affiliations:** 1Department of Geological Sciences, California State University Long Beach, 1250 Bellflower Boulevard, Long Beach, CA 90840, USA; 2Silixa LLC, 16203 Park Row, Suite 185, Houston, TX 77084, USA; thomas.coleman@silixa.com

**Keywords:** distributed acoustic sensing, fiber optic sensors, earth tides, low frequency strain, geomechanics

## Abstract

The solid Earth strains in response to the gravitational pull from the Moon, Sun, and other planetary bodies. Measuring the flexure of geologic material in response to these Earth tides provides information about the geomechanical properties of rock and sediment. Such measurements are particularly useful for understanding dilation of faults and fractures in competent rock. A new approach to measuring earth tides using fiber optic distributed acoustic sensing (DAS) is presented here. DAS was originally designed to record acoustic vibration through the measurement of dynamic strain on a fiber optic cable. Here, laboratory experiments demonstrate that oscillating strain can be measured with DAS in the microHertz frequency range, corresponding to half-day (M_2_) lunar tidal cycles. Although the magnitude of strain measured in the laboratory is larger than what would be expected due to earth tides, a clear signal at half-day period was extracted from the data. With the increased signal-to-noise expected from quiet field applications and improvements to DAS using engineered fiber, earth tides could potentially be measured in deep boreholes with DAS. Because of the distributed nature of the sensor (0.25 m measurement interval over kilometres), fractures could be simultaneously located and evaluated. Such measurements would provide valuable information regarding the placement and stiffness of open fractures in bedrock. Characterization of bedrock fractures is an important goal for multiple subsurface operations such as petroleum extraction, geothermal energy recovery, and geologic carbon sequestration.

## 1. Introduction

The motions induced in the solid Earth through tidal forces are known as earth tides. Tidal forces exerted on the Earth are primarily from the gravitational pull of the Sun and Moon. Gravitational acceleration is, therefore, not uniform about the Earth in either space or time. Most major tidal waves are exerted either diurnally or semi-diurnally, although longer periods occur also (Table 1). The semi-diurnal lunar earth tide, M_2_, causes a vertical displacement of 632 mm at the equator, for example (Table 1). Earth tides have been measured primarily using surface tiltmeters and borehole strain meters [1]. Earth tides compress porous earth which can express water to deep boreholes. The change in water levels in boreholes and wells in response to earth tides have been often used to estimate geologic compressibility and permeability when combined with additional geologic information [2,3,4,5]. Volume strain affects the pore-pressure response which, in turn, affects the water-level response in the well. 

Earth-tide driven water level changes have been particularly useful for characterizing the permeability of fractures in bedrock. Information about the geomechanical properties and the orientation of the fractures can be gleaned from water level data [2,3,5]. Bower [2], for example, developed equations that relate tidal strain to fracture aperture displacement for assumed orientations of fractures. However, well-water levels integrate the response to strain over the hydraulically isolated interval so it is not possible to isolate particular fractures or fracture sets within the conductive horizon. Multi-level wells increase the number of zones that can measure the integrated hydraulic response; however, the maximum number of monitoring intervals is limited. Extensiometers can be deployed in boreholes to measure only one fracture or set of fractures [2,4,6]. These devices are very accurate and sensitive, but they are fragile, difficult to deploy, and can measure only intervals which have been identified a priori through other logging techniques.

Here we introduce a new concept for sensing earth-tide measurement based upon fiber optic distributed acoustic sensing (DAS). DAS is typically used to measure strains in frequencies of Hz or kHz. In previous work we have demonstrated in the laboratory [7] and field [8,9,10] that frequencies in the range of mHz can be extracted from DAS signals. The current study focuses on laboratory studies in which we show that periodic strains can be extracted in the microHertz (μHz) range, i.e., a frequency of 23.1 (μHz) or a period of 0.5 day. The purpose was to demonstrate that ultra-low frequency signals can be extracted from DAS, with the intention of supporting future deep borehole measurements of earth-tide strain using DAS. To our knowledge, this is the lowest isolated frequency DAS signal that has been reported.

It is important to recognize that the native measurement of DAS can be strain rate, not strain as is measured in extensometers. This requires that a dynamic (periodic) strain be measured and that the lower the frequency, the poorer the signal to noise ratio. Isolation of ultra-low-frequencies, therefore, is expected to be challenging for earth tides. However, our success in the mHz frequency suggested earth-tide measurements might be possible. We have measured, for example, periodic fracture displacements in a rock borehole of less than 0.6 nm at a frequency of ~1 mHz [10] and displacement of fiber in the laboratory of less than 1 pm at a frequency of ~10 mHz [7].

## 2. Materials and Methods

The use of DAS for monitoring borehole seismic and acoustic signals has migrated from early field demonstration to commercial use within a span of 5 years [12,13,14,15,16]. The general principals of DAS are explained in the review by Miah and Potter [17] and the text by Hartog [18].

### 2.1. Sensor Principles

The instrument used for these tests was a Silixa iDAS^TM^ system (Silixa Ltd., Elstree, UK). Like all DAS systems, the iDAS consists of an interrogator unit attached to a fiber optic (FO) cable. The instrument takes advantage of the coherent nature of Rayleigh backscatter from a narrowband (laser) light source [18,19]. If strain is applied at a point along the optical fiber, scattered signals upstream and downstream stream of that point are shifted in phase from one another, with a magnitude linearly related to the displacement in the optical fiber. In differential phase DAS (dΦ − DAS), backscatter phase is compared between sections upstream and downstream of the measurement channel location, separated by a “gauge length”. If u(z,t) is the dynamic displacement of the fiber at the interrogation location, *z*, at time, *t*, the DAS output is an estimate of: (1)$$\left[u\left(z+\frac{dz}{2}\;,t+dt\right)-u\left(z-\frac{dz}{2}\;,t+dt\right)\right]-\left[u\left(z+\frac{dz}{2}\;,t\right)-u\left(z-\frac{dz}{2}\;,t\right)\right].$$

The gauge length and temporal sampling interval are represented in (1) by *dz* and *dt*, respectively [20]. The iDAS output is an estimate of fiber displacement rate:(2)$$\frac{\partial }{\partial t}\left(\frac{\partial u}{\partial z}\right).$$

An integrated displacement may also be obtained by changing Equation (1) from a difference to a summation,
(3)$$\int \frac{\partial u}{\partial t}\;dt,$$
but this scheme presents ranging problems for signal processing, so is not typically used by the iDAS instrument. For the measurements discussed here, a gauge length (*dz*) of 10 m and a spatial sampling interval (measurement channel spacing) of 0.25 m was used. A temporal sampling frequency (1/*dt*) of 1 kHz was used. When the time differential approach is used (2), signal to noise is sensitive to the time rate of change in strain. Better signals will be received from the instrument for greater rates of strain, until the Nyquist frequency of the temporal sampling rate is approached. 

### 2.2. Experimental Setup

Periodic strain was applied to the FO cable using hydraulics. A 64 m length of tight-buffered optical fiber) was wrapped around a 10 cm diameter rubber bladder which was slid over a 9 cm diameter perforated cylinder (Figure 1). This assembly was submerged in a 15 cm diameter cylindrical reservoir of water. The inner cylinder was filled with water about 10 cm over the water level in the outer cylinder. An epoxy-filled 6.4 cm diameter PVC pipe “slug” was used to displace the water in the inner cylinder which, in turn, caused the bladder to expand and strain the wrapped fiber. The slug was attached to a rotating disc that caused it to move up and down in a sinusoidal displacement. Pressure was measured in the inner cylinder with a self-contained logger (Levelogger Jr, Solinst Ltd., Georgetown, Ontario, Canada) and outside with a high-precision pressure transducer vented to the atmosphere to correct for barometric pressure (PX409-005G USBH, Omega Engineering Inc., Norwalk, CT, USA). Unfortunately, the transducer appeared to be improperly vented and drifted badly during the three-day experiment. As a consequence, a short term 10-minute period experiment was run afterward to relate the pressure in the inner and outer cylinders. The amplitude of displaced water in the outer cylinder was 0.28 mm and showed no delay from the water level change in the inner cylinder. Based upon the dimensions of the submerged cylinder/bladder, strain in the fiber was estimated to be approximately 0.07% suggesting a displacement amplitude of 7 mm over the 10 m DAS gauge length. These volumetric calculations are very rough, however, given the uncertainties in the head measurements and bladder geometry. 

### 2.3. Data Processing

Raw DAS signals were converted to displacement rate (nm/s). The mean response over a specified number of channels was calculated for each file and then concatenated until a single file held the mean response over the three-day experiment duration. To obtain frequency domain plots, a Fast Fourier Transform (FFT) was applied to the data to find ultra-low-frequency signals. Because data were collected at 1 kHz sampling rate and the signal of interest was at 23.1 μHz (1/2 day period), only a very small fraction of the frequency spectrum was used. The amplitude of displacement of the fiber due to the hydraulic oscillation was calculated by dividing the amplitude of the displacement rate by the frequency at the FFT peak. 

Plots of time domain displacement were obtained by integrating the displacement rate (nm/s) through time using the “cumtrapz” function in Matlab™. The sign of strain rate was erroneously reversed in the output file so was corrected after integration. A small bias in the displacement rate data caused the integrated signal to drift about 2000 nm/day. The drift was removed through a high-pass Butterworth filter, with a pass band of half the frequency of the diurnal peak. Because integration is essentially a low pass filter, the DAS data are band-pass filtered to obtain a displacement time series. 

## 3. Results

The ultra-low frequency end of the FFT spectrum is shown in Figure 2. The square denotes the frequency corresponding to a 12-h period (23.1 μHz). A clear local peak appears at this hydraulically induced frequency, with two secondary peak frequency multiples (denoted by diamonds) of 2 (6-h period) and 3 (4-h period). The displacement rate amplitude is 38 nm/sec which, for a perfect sine, converts to a displacement amplitude of 1.7 mm measured over the 10 m gauge length, or 165,000 nanostrain. The first multiple is at the same approximate phase as the primary frequency but the second multiple is shifted by about 6 h from the primary frequency. 

The mean DAS displacement is compared with head measured in the inner cylinder in Figure 3. It would have been better to compare to head measurements in the outer cylinder, where water was displaced by the expanding bladder, but as mentioned previously, this transducer failed to capture the head change over the three-day period. However, since no lag was observed between the head recorded internally and the head recorded externally to the bladder, the strain indicated by the heads in the cylinder are synchronous with the strain measured by DAS. 

The displacement shown in Figure 3 is the result of taking the mean value of displacement rate over the entire length of the fiber around the flexible bladder (160 channels). This represents a response stacking that improves signal to noise (S/N). To evaluate the effect of stacking on S/N, we compare the FFT amplitude responses for the mean response in 160, 80, and 40, channels (Figure 4). These correspond to 40, 20, and 10 m fiber lengths of stacked response. Recall, that displacement is measured over a gauge length of 10 m, so that the displacement recorded in less than 40 channels still represents integrated displacement in 10 m of fiber. Figure 4 demonstrates that there is no improvement of signal beyond stacking 80 channels (20 m) at the earth-tide frequency (23.1 μHz). The multiple frequency detections persist in spite of the signal stacking. 

The DAS instrument was able to sense periodic displacements along the fiber optic cable to frequencies that correspond to a period of one-half day. Because the M_2_, N_2_, and S_2_ tidal species are all at periods of approximately one-half day (Table 1), there appears to be potential for measuring earth-tide strain in boreholes using current DAS technology. However, although the frequencies measured in these tests are comparable to earth tides, the amplitudes are not. Our laboratory tests were designed to maximize strain because it was not known if sub-mHz frequencies could be measured by DAS at any amplitude. While the strain induced in the laboratory tests were about 165,000 nanostrain, earth-tide strains measured in 12 borehole strainmeters in California, for comparison, were between one and 23 nanostrain [21]. A key question remains: are earth-tide strain magnitudes measurable in the field by DAS?

Practical measurement of earth tides in DAS depend upon the signal-to-noise (S/N) ratio of the recorded displacement rate. There are a number of ways to enhance S/N. First, deployment in a borehole will reduce environmental acoustic noise considerably. These experiments were carried out in the third floor of a building that is subject to variation in barometric and building air pressures. Some of the harmonics may have been due to the movement of the slug in the cylinder which was inherently noisy because of the mechanics of the rotating disc. In a borehole, only barometric, seismic, ocean tide, and earth tides should produce frequencies of strain in the range of interest.

Stacking was shown to improve S/N in these tests (Figure 4). Because Earth-tide forces have enormously large wavelengths, strain is distributed over large Earth volumes. It is likely that channels will be stacked in interpretation of the subsurface strain, although the number of channels is a function of the desire to localize strain behavior. For example, if the objective is to measure strain in a 1-m thick fracture zone, then only four channels should be stacked. However, if the objective is to measure strain contrasts between formations of 50 m thickness, then 200 channels can be stacked.

Finally, DAS instrumentation continues to undergo improvement. The instrument used for this experiment has a S/N ratio 100 times poorer than the latest instrument/specialty fiber (Carina^®^/Constellation™, Silixa Ltd., Elstree, UK) from the same vendor [22]. We anticipate that technology will continue to improve and be deployed for oil and gas, carbon sequestration, and geothermal applications. These installations could also be tested for earth tide frequencies to enable fracture geomechanical and hydromechanical characterization, assuming that data are collected over a period of several days or more.

## 4. Conclusions

These laboratory experiments demonstrate that DAS is capable of measuring strain frequencies that correspond to semi-diurnal earth tides. The sensitivity to strain magnitude has yet to be demonstrated, however, as that will likely need to be done in boreholes, to exclude barometric and other noise sources within the same frequency range. In addition, instrumentation is now commercially available that would significantly improve the signal to noise ratios and associated strain magnitude detection limit of these measurements. These improved systems makes use of a special interrogator paired with custom-engineered optical fiber, but increases the signal to noise by two orders of magnitude. As DAS instrumentation improves, it is likely that earth tides will be measurable in boreholes installed with fiber optic cable.

Most DAS instruments are not ideally suited to measuring earth tides because their native measurement is displacement rate or strain rate, rather than strain. Consequently, the sensitivity of DAS to strain measurements degrades with lower frequencies. Earth tides currently represent the extreme limit of DAS strain sensitivity. However, DAS has an enormous advantage over borehole extensiometers and tilt meters, and even the response of fluid pressure in wells. DAS produces a distributed measurement of strain along the length of the fiber optic cable. This allows the depth localization (logging) of geomechanical response to earth-tide forcing. Response in fault zones, for example, could be measured without knowing the depth of a fault, a priori.

The use of earth tides has been suggested to aid characterization and monitoring reservoirs of groundwater [3], oil and gas [23], geothermal heat [24], and CO_2_ sequestration [11]. Should DAS monitoring become practical at Earth tide frequencies, multiple opportunities exist for applications of this technology. When combined with more traditional Earth tide monitoring approaches such as tiltmeters, extensiometers, and well pressure response, the data could be more definitive for reservoir investigations. Field testing in a borehole would provide an important evaluation of DAS technology toward practical Earth-tide applications.

## Figures and Tables

**Figure 1 sensors-19-01975-f001:**
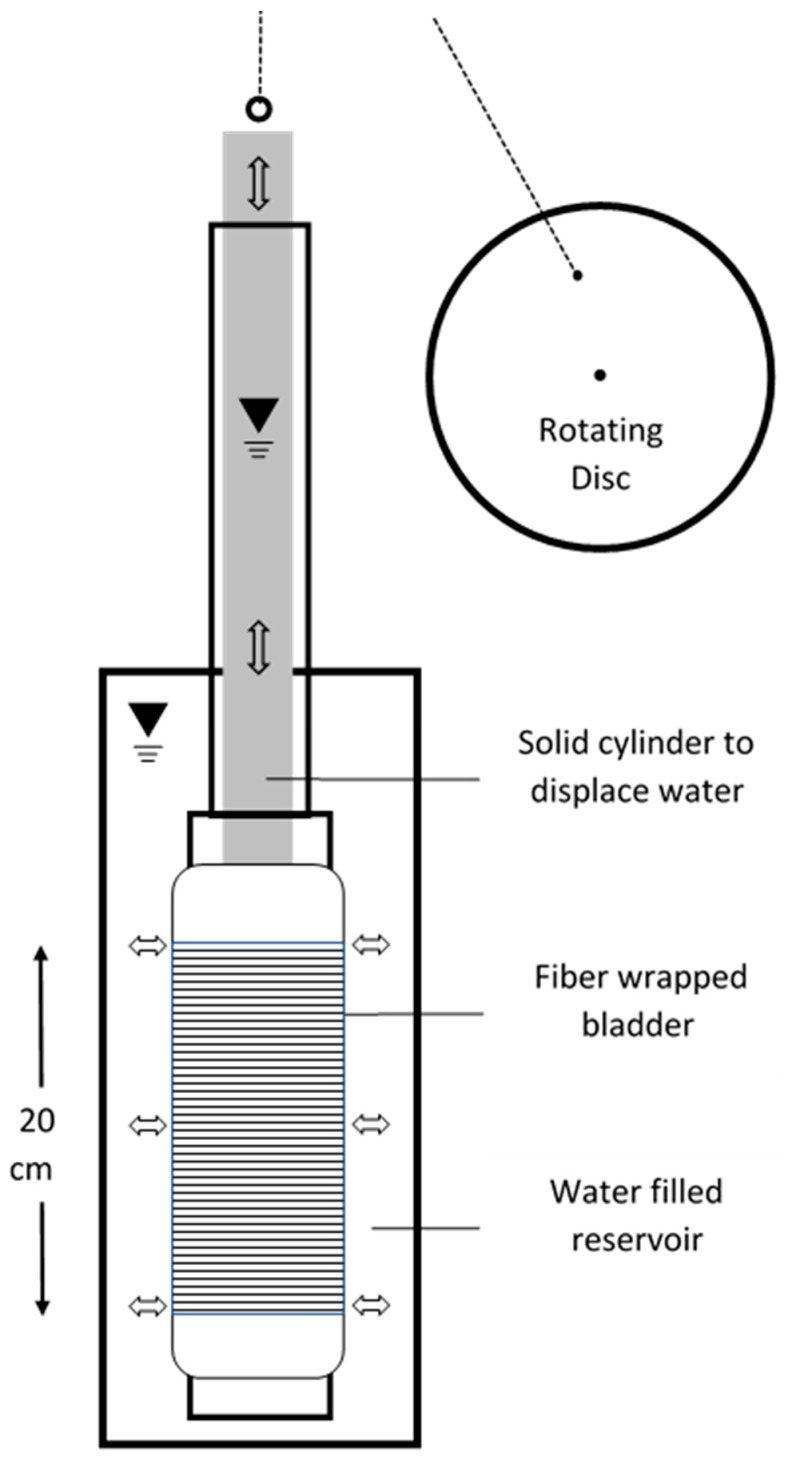
Schematic of the experimental setup.

**Figure 2 sensors-19-01975-f002:**
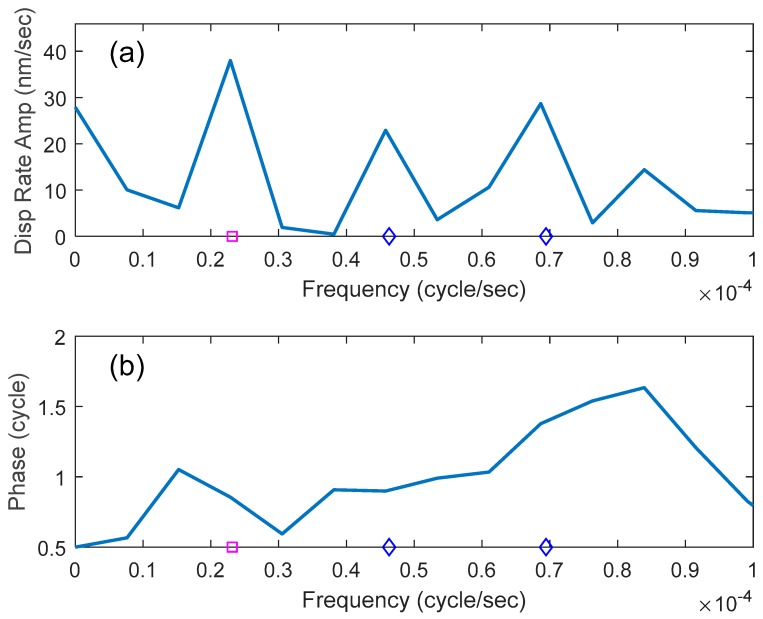
Fast Fourier Transform spectra of mean displacement of 160 channels (360–519). The pink square represents a frequency corresponding to a 12-h period, and the blue diamonds indicate multiple frequencies corresponding to 6 and 4 h periods, respectively, (**a**) amplitude spectrum, (**b**) phase spectrum.

**Figure 3 sensors-19-01975-f003:**
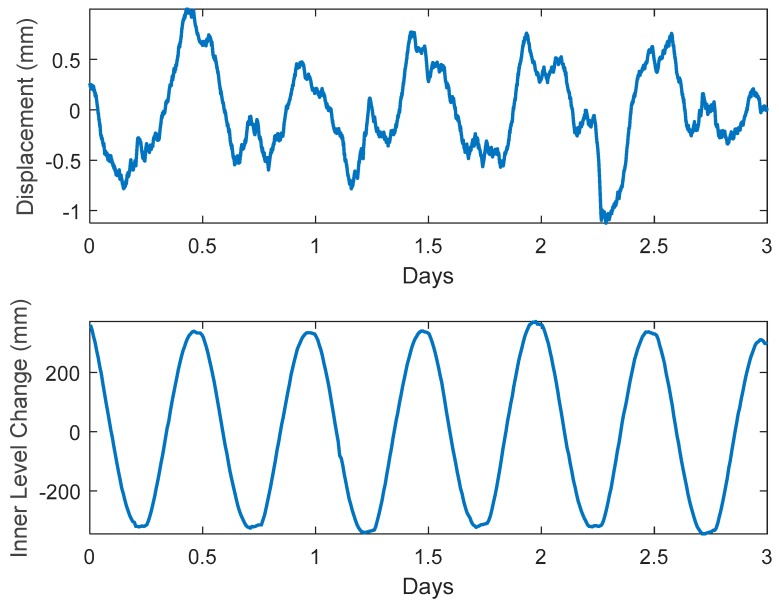
DAS measured displacement (**top**) compared with water level change inside the bladder, that induces strain in the fiber (**bottom**).

**Figure 4 sensors-19-01975-f004:**
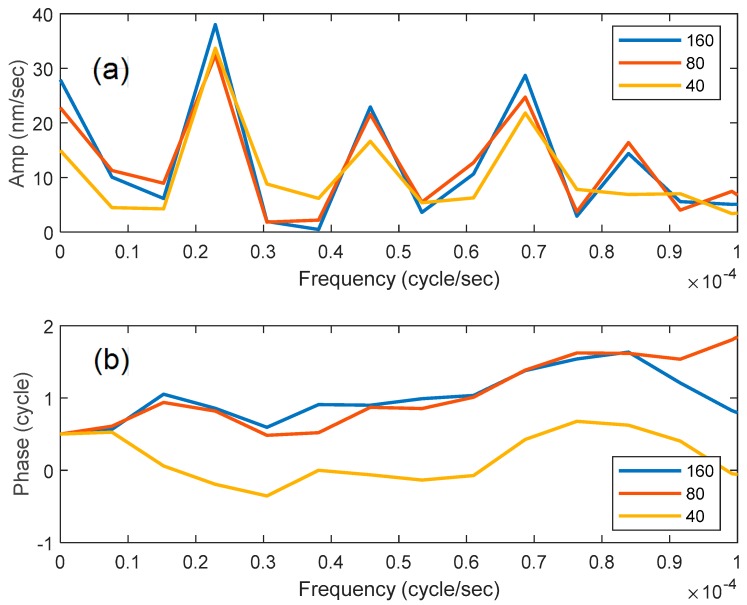
FFT spectra of displacement rate as a function of signal stacking: 40, 80, and 160 channels, (**a**) amplitude spectrum, (**b**) phase spectrum.

**Table 1 sensors-19-01975-t001:** Major earth-tide waves at the equator [11].

Symbol	Origin	Period (day)	Amplitude (mm)
Long-period components			
SS_a_	Solar declinational	182.6	31
M_m_	Lunar elliptic	27.6	35
M_f_	Lunar declinational	13.7	67
Diurnal components			
O_1_	Lunar principal	1.08	262
P_1_	Solar principal	1.00	122
mK_1_ + sK_1_	Lunar-solar declinational	1.00	369
Semi-diurnal components			
N_2_	Lunar major elliptic of M2	0.527	121
**M_2_**	**Lunar principal**	**0.518**	**632**
S_2_	Solar principal	0.500	294

## Data Availability

Mean dynamic strain data (nm/s) for channels 360–519 are available on the Geothermal Data Repository: California State University. (2019). Distributed Acoustic Sensing of Strain at Earth Tide Frequencies [data set]. Retrieved from http://gdr.openei.org/submissions/1129.

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
