# Peer review of "Distributed Acoustic Sensing of Strain at Earth Tide Frequencies"

_sensors, 2019, doi:10.3390/s19091975_

Reviewer 1 Report

In line 41, it is written geomechnical, which likes to be wrongly written.

In line 72, It is introduced an iDAS system. However, it is not clear if iDAS is the same as DAS. 

In line 101, it is described the optical fiber cable (900um, single mode Corning). The 900um dimension what is denoting?. If it denoted de diameter of the fiber, it looks to be so big for a standard single mode fiber. Please give more detail about the OF cable.

A picture of your experimental setup would be very illustrative for the readers. In order to improve the purpose of Fig. 5

In line 170, you are talking about periods of approximately one-half day. However, in table 1 the periods which are you talking about are around one day. 

Author Response

In line 41, it is written geomechnical, which likes to be wrongly written.

Corrected

In line 72, It is introduced an iDAS system. However, it is not clear if iDAS is the same as DAS. 

Thank you, we have removed this sentence and address the specific instrument in the following paragraph.

In line 101, it is described the optical fiber cable (900um, single mode Corning). The 900um dimension what is denoting?. If it denoted de diameter of the fiber, it looks to be so big for a standard single mode fiber. Please give more detail about the OF cable.

We have clarified by specifying:  900 um OD buffered, 9/125 um singlemode fiber, Corning

A picture of your experimental setup would be very illustrative for the readers. In order to improve the purpose of Fig. 5

Because you cannot see the fiber or the oscillating cylinder, the photograph is not very helpful so we are sticking with the drawing.

In line 170, you are talking about periods of approximately one-half day. However, in table 1 the periods which are you talking about are around one day. 

A typo, thank you for correcting. We now write:  Because the M2, N2, and S2 tidal species are all at periods of approximately one-half day (Table 1), there appears to be potential for measuring earth tide strain in boreholes using current DAS technology

Reviewer 2 Report

The application of DAS to observe earth tides is challenging, because of the ultra-low frequency of tides and the difficulty in identifying effective signals from noises. In this paper, the authors illustrated the feasibility of DAS in looking at earth tides and also the data analysis method. This paper is well organized and properly presented. One question is, is there any other supporting method or data to confirm the signal obtained by DAS is related to earth tides? Only the ultra-low frequency is not enough. Also, Figure 1 should be better illustrated including the sizes. Finally, I am looking forward to the in-situ results.

Author Response

The application of DAS to observe earth tides is challenging, because of the ultra-low frequency of tides and the difficulty in identifying effective signals from noises. In this paper, the authors illustrated the feasibility of DAS in looking at earth tides and also the data analysis method. This paper is well organized and properly presented.

One question is, is there any other supporting method or data to confirm the signal obtained by DAS is related to earth tides? Only the ultra-low frequency is not enough.

As stated in the manuscripts, these are only laboratory tests that show that ultra-low frequencies can be obtained by the instrument.  We discuss the relevance to earth tides in the conclusions.  It is not clear what modifications the reviewer is requesting.

Also, Figure 1 should be better illustrated including the sizes.

We have revised the figure to include a scale

Finally, I am looking forward to the in-situ results.

We also.